# Perception of Velocity during Free-Weight Exercises: Difference between Back Squat and Bench Press

**DOI:** 10.3390/jfmk7020034

**Published:** 2022-04-18

**Authors:** Ruggero Romagnoli, Maria Francesca Piacentini

**Affiliations:** 1Departement of Human Movement and Health Sciences, University of Rome “Foro Italico” Piazza L. De Bosis 15, 00135 Rome, Italy; r.romagnoli2@studenti.uniroma4.it; 2Italian Weightlifting Federation ‘FIPE’, 00135 Rome, Italy; 3Department of Human Physiology and Sports Medicine, Universiteit Brussel, 1030 Brussels, Belgium

**Keywords:** resistance training, autoregulation, velocity-based training, monitoring intensity, linear position transducer

## Abstract

The perception of bar velocity (PV) is a subjective parameter useful in estimating velocity during resistance training. The aim of this study was to investigate if the PV can be improved through specific training sessions, if it differs between the back squat (SQ) and bench press (BP), and if there are differences in perception accuracy in the different intensity zones. Resistance-trained participants were randomly divided in an experimental (EG, n = 16) or a control group (CG, n = 14). After a familiarization trial, both groups were tested before and after 5 weeks of training. The PV was assessed with five blinded loads covering different intensity domains. During the training period, only the EG group received velocity feedback for each repetition. Prior to training, both groups showed a greater PV accuracy in the SQ than in the BP. Post training, the EG showed a significant reduction (*p* < 0.05) in the delta score (the difference between the real and perceived velocity) for both exercises, while no significant differences were observed in the CG. Prior to training, the perceived velocity was more accurate at higher loads for both exercises, while no difference between loads was observed after training (EG). The results of this study demonstrate that the PV improves with specific training and that differences in the accuracy between loads and exercise modes seen prior to training are leveled off after training.

## 1. Introduction

Both recreational and competitive athletes include resistance training in their program, with the precise scope to maximize strength, power, hypertrophy, and local muscular endurance [1,2]. RT programs are designed to achieve these specific objectives by manipulating different variables such as: muscle actions, resistance, volume (total number of sets and repetitions), exercises selected and workout structure (e.g., the number of muscle groups trained), the sequence of exercise performance, rest intervals between sets, repetition velocity, and training frequency [2]. Normally, the most utilized variables are intensity, volume, and frequency. RT intensity is commonly identified with the relative load lifted (% of 1RM). This method, called percent-based training (PBT), does not take into account the effects of the inter-set recovery periods or the number of repetitions performed in each set at a given load [3,4]. However, due to daily fluctuations in 1RM and the necessity to continuously test for 1RM in order to prescribe correct training intensities [3], most recently, different studies reported the benefits of prescribing RT via the autoregulation methodology. The most used autoregulatory approaches to resistance training include subjective methods such as the rating of the perceived exertion (RPE) scale or the repetitions in reserve (RIR) method and objective methods such as velocity-based training (VBT) {Formatting Citation}. VBT, in fact, has been extensively studied as a way of monitoring exercise intensity [5,6,7,8,9] and allows to estimate the % 1RM from the actual velocity of each repetition, without performing demanding maximal tests to adjust the training loads. This method allows estimating the daily readiness (or daily 1RM) and monitors the decrease in velocity within each set to manage the accumulation of fatigue, thus being effective for improving strength levels and performance parameters such as sprints and jumps [10,11,12]. However, the VBT methodologies require velocity measuring devices in order to correctly prescribe training and be sure athletes are in the correct training zone. These devices, although more affordable than a few years ago, may still be prohibitive in terms of the costs for many coaches and teams [13]. Moreover, in sports with many athletes training in the gym at the same time, not every member of the team may dispose of the measuring device during their training session.

The perception of bar velocity (PV) is a subjective parameter and has been shown to be valid in estimating real velocity during the back squat and bench press [14,15,16]. The authors validated a velocity perception scale ranging from 0.1 to 1.6 m·s^−1^ that is accompanied by five qualitative descriptors (e.g., “very slow” and “power zone”). Only one familiarization stage and two validation stages were utilized to achieve a high accuracy of prediction. However, the perceived execution velocity was overestimated at lower %1RM while underestimated at higher intensities. A longer process of familiarization with the use of the scale may probably be required because, to date, it is not known if the perception can be further improved considering that a variation of the Vr of 0.07–0.09 m·s^−1^ corresponds to a variation of 5% of the relative load (% 1RM) [17]. Moreover, in view of the differences between the BP and SQ in the range of motion, muscle mass, and body position, a different accuracy in the perception of velocity might be possible between the two exercises.

Therefore, the aim of the present study was threefold, specifically to evaluate:(1)If the PV can be improved through specific training sessions twice a week for 5 weeks;(2)If the PV differs between exercise modalities (upper or lower body exercise, supine and upright position);(3)If the PV accuracy differs in the different intensity zones.

The main hypothesis was that there would be a training effect and no difference between the exercise modalities but that the power zone (the zone more targeted with the VBT) would be the most difficult to perceive accurately.

## 2. Materials and Methods

### 2.1. Study Design

A longitudinal pre–post design was used to compare the effect of the training protocol on perception velocity at different loads. All participants underwent one familiarization session and four test sessions, including two 1RM test sessions for SQ and BP (in a randomized order) and two PV test sessions. The 1RM tests were performed on different days separated by at least 48–72 h. The first PV session was performed 48–72 h after the last 1RM test, the second after 5 weeks. Participants were randomly assigned to an experimental (EG) or a control (CG) group and were required to avoid any 1RM tests or strenuous exercise in the two days prior to the assessments.

### 2.2. Subjects

Thirty resistance-trained participants (males, n = 21; females, n = 9; age= 26.2 ± 4.1 years; body mass = 74.5 ± 14.1 kg; height= 177.1 ± 9.4 cm) were randomly divided in the experimental group (EG, n = 16, 12 males and 4 females) or the control group (CG, n = 14, 9 males and 5 females). Subjects’ characteristics and 1RM are presented in Table 1.

Inclusion criteria were at least two years of experience in resistance training, no previous experience in VBT, no muscle or bone injury before or during the intervention period, familiar with back squat and bench press exercises. All participants received detailed information regarding the procedures and signed a written informed consent. The study protocol adhered to the Declaration of Helsinki and was approved by the Institutional Review Board (CAR. 75/2021).

### 2.3. Procedures

#### 2.3.1. The 1RM Test Sessions

Prior to maximal tests assessment, subjects participated in a familiarization session devoted to standardizing the execution of each exercise. All participants completed two 1RM tests for bench press (BP) and back squat (SQ) using free weights.

The execution required in the SQ involved a depth such that, viewed from the lateral aspect, the crease of the hip was below the top of the knee. In the BP, participants had to maintain contact points at all times of the lift (head, upper back, buttocks, and flat feet). Each test session began with a general dynamic warm-up consisting of bodyweight exercises, mobility, and dynamic stretching followed by a specific warm-up consisting of sets with progressive loading. After the warm-up, 5 repetitions were performed with an initial load of 20 kg; subsequently, the number of repetitions and the load increases were adjusted according to the mean propulsive velocity achieved so that 1RM could be accurately determined. A detailed description of the protocol has recently been reported elsewhere and is shown in Table 2 [18]. Mean propulsive velocity of the barbell that we will consider the real velocity (Vr) was recorded for each repetition by a linear position transducer (LPT) (Vitruve, SPEED4LIFTS S.L., Madrid, Spain).

These tests were used as familiarization for the perception of velocity scale (PV). During the 1RM tests, subjects were provided with the minimum and maximum velocity reached during each set and were asked to visualize these velocities on the scale. All sessions were supervised by a Certified Strength and Conditioning Specialist (CSCS—NSCA) and 3 spotters were present to ensure safety and proper performance.

#### 2.3.2. Perception Assessment

Before (Pre) and after (Post) the training period, the accuracy of the PV was assessed with 5 blinded loads in a random order for both groups. The loads were chosen based on the velocities reached during the 1RM-tests, in order to cover a wide range of intensities, as shown in Table 3. For each load, the athletes were asked to perform 2 repetitions at maximum velocity. At the end of the set, they had to report the fastest and their PV (Vp) using the velocity scale (Bautista et al.) [14].

#### 2.3.3. Training Program

The EG group followed 5 weeks of specific training (10 sessions, 2 days/week). During the 5 weeks of training, they received visual and auditory feedback (with the LPT) while lifting different loads and viewed the scale at the end of each set.

In the training protocol, described in detail in Table 4, we prescribed sets, repetitions, and repetitions in reserve (RIR); therefore, the load was chosen by the subjects, based on RIR. In the first 2 sets (warm-up), they had to complete the prescribed repetitions with 2–3 RIR, the following sets had to be completed with 1 RIR. The weekly schedule consisted of 2 sessions including back squat and bench press. On the first day, the SQ was performed with heavy loads and the BP with light loads. On the second day, the exercises were reversed (BP: heavy, SQ: light) to ensure that within the week, the 2 exercises were performed with the same modalities (same volume of training, alternating heavy, medium, and light loads). The CG group continued training for 5 weeks, 2–3 times a week, including SQ and BP, but without feedback on velocity.

### 2.4. Statistical Analysis

The normal distribution of the data was assessed by the Shapiro–Wilk test. Pearson’s Chi-Square test was conducted for qualitative variables. Bland–Altman plots were constructed to explore the agreement between variables (Vr, Vp). Delta score (ds) was calculated as follows: Vp–Vr. Friedman test and Wilcoxon signed-rank tests were used to identify any differences in ds between exercises and loads. Quantitative variables are reported as median ± interquartile range and qualitative variables as percentage. The established significance level was *p* ≤ 0.05. Statistical analyses were performed in SPSS v25 (SPSS Inc., Chicago, IL, USA).

## 3. Results

### 3.1. Qualitative Analysis

A qualitative analysis was conducted to investigate whether the subjects were able to discriminate the fastest between the two repetitions during the blinded load test. The EG group correctly identified the fastest repetition 55% of the time for the SQ and 59% for the BP in Pre, and 76% for the SQ (*p* < 0.05) and 71% for the BP in Post (*p* = 0.097). The CG group did not improve the identification of the fastest repetition (64% for the SQ and 60% for the BP in Pre, and 56% (SQ, *p* = 0.301) and 57% (BP, *p* = 0.731) in Post).

### 3.2. Agreement between Perceived and Real Velocity

Figure 1 reports the Bland–Altman plots to describe the agreement between the real (Vr) and perceived (Vp) velocities. The bias is indicated with a thick solid line and the limits of agreement (ranged between bias − 1.96 × SD and bias + 1.96 × SD) with dotted lines.

The EG group showed a clear improvement in the Post condition, with a reduction in the bias and limits of agreement from 0.038 (−0.441–0.516) m·s^−1^ in the SQ Pre to −0.025 (−0.250–0.199) m·s^−1^ in the SQ Post and from 0.206 (−0.215–0.626) m·s^−1^ in the BP Pre to 0.082 (−0.183–0.347) m·s^−1^ in the BP Post. In the CG group, no differences in the Pre and Post conditions were found.

### 3.3. Delta Score (ds)

Table 5 summarizes the ds, calculated as Vp–Vr, for the EG and the CG for each condition (SQ Pre, SQ Post, BP pre, BP Post) and the results are presented as the median ± interquartile range. The delta score represents the deviation of the perceived velocity (Vp) from the real velocity (Vr) measured with an encoder. The Friedman test confirmed that the delta score was different across the exercises and loads in both groups. The Wilcoxon tests showed for the EG a significant reduction in the ds for the BP between Pre and Post (*p* < 0.05) and a non-statistically significant reduction between the SQ Pre and SQ Post (*p* = 0.057). In the CG, the ds increased from Pre to Post, both for the SQ and for the BP, with no significant differences (*p* = 0.06, *p* = 0.323, respectively).

Figure 2 summarizes the ds for the GC and CG for each condition and each load. The loads are presented in ascending order from the lightest (load 1) to the heaviest (load 5).

**EG:** In the SQ, the delta score between the Pre and Post conditions decreased significantly (*p* < 0.05) in load 1, load 2, load 4, and load 5, and a non-significant reduction was found in load 3 (*p* = 0.743), while in the BP, significant reductions (*p* < 0.05) were found in loads 1, 2, and 3 but not in load 4 (*p* = 0.160) and in load 5 (*p* = 0.651).

**CG:** Both in the SQ and in the BP, the delta score worsened between the Pre and Post conditions with significant increases (*p* < 0.05) in load 2, while for loads 1, 3, 4, and 5, no significant differences were found (SQ: *p* = 0.718, *p* = 0.899, *p* = 0.087, *p* = 0.99; BP: *p* = 0.432, *p* = 0.782, *p* = 0.933, *p* = 0.164, respectively).

## 4. Discussion

The main findings of this study were that 5 weeks of specific training utilizing the LTP and the perception of velocity scale increased the accuracy of the perception of the barbell velocity in the EG. Before training, the qualitative analysis revealed that for only 55% of the time athletes were able to discriminate the fastest repetition within a set of two for the SQ exercise, while after training, the accuracy arrived at 79%. For the BP, the same results were observed. Moreover, the delta score (i.e., the difference between the perceived and real velocity) decreased significantly in the EG, indicating a more precise perception of velocity, while no differences were seen in the CG. Considering that VBT has been shown to be a reliable and effective methodology to prescribe resistance training in different sport populations [9], the use of the barbell velocity to estimate the intensity of the lift based on the linear relationship between the load and velocity is of great interest to correctly prescribe training. Although technology has greatly improved and small and portable devices are available on the market to measure the barbell velocity, the cost still might be impractical for many coaches. Moreover, when training or testing a group of athletes, multiple devices would be necessary and therefore challenging. For this reason, recent research has focused on validating a velocity perception scale during bench press [14] and squat exercises [15]. In the first study [14], subjects were tested during 5 days over a spectrum of intensities ranging from light, medium, and heavy, and the authors found a positive correlation between the perceived and real velocity. The second study [15] instead tested the concurrent validity of the perceived velocity at intensities ranging from 20 to 70% 1RM during the full back squat. However, the BP and SQ exercises were never compared directly.

Because bodily self-consciousness is influenced by a visual, vestibular, and direction of gravity that are different depending on the body position (standing upright, sitting on a chair, and lying supine) [19,20], the same athlete could perceive differently (and with different accuracy) the barbell velocity during the SQ or BP. During the BP exercise, the athlete is lying supine and moves the barbell, while during the SQ, the athlete stands upright and moves together with the barbell. Moreover, not only differences regarding body position but also the range of motion and muscle mass involved could ultimately contribute to possible differences in the accuracy of the perception of velocity of the barbell in the same group of subjects. Only one study directly compared the SQ and BP, although monitoring perceived changes in velocity during a set [16]. They reported that the odds of underestimating the change in velocity were 4-fold higher in the SQ compared with the BP and that the accuracy increased in the BP with sets and repetitions. They hypothesized that this difference could be attributable to the shorter distance the barbell travels during the BP, to the fact that participants can visually observe the movement of the barbell during the BP and that upper body muscles are more used during daily activities specifically for fine movements, probably increasing the accuracy of perception.

Although we performed an acute study on selected loads (and not sets), we found more accuracy in the SQ compared to the BP for both groups in the Pre condition. While the CG also continued to show a better perception of velocity during the SQ in the Post test, the EG, with training, improved the PV in both the SQ and BP, making these exercises very similar in terms of accuracy. These discordant results could probably be explained by the different strength level between participants. In fact, the SQ exercise showed similar values between studies, while the BP data seem to indicate that their participants were better performers. This could be due to more training with this specific exercise, explaining the better accuracy they found in the perception of the velocity in the exercise where subjects show a greater level of expertise.

We opted for only one familiarization test prior to the Pre test session as it was previously reported to be sufficient [14,15]. In fact, no differences were observed between the groups in the Pre tests. However, the EG after 5 weeks of training significantly improved the PV in both the SQ and BP at almost all loads, while the CG not only did not improve but reported that they could not remember the velocity ranges on the scale and how to correctly perceive the barbell velocity.

In order to understand the accuracy in perceiving the barbell velocity, we utilized the delta score that calculates the difference between the perceived (Vp) and the real (Vr) velocity. We chose to calculate the delta score parameter to show how much the perception of the velocity value (Vp) of the subjects deviated from the real value (Vr). Since a variation of the Vr of 0.07–0.09 m·s^−1^ corresponds to a variation of 5% of the relative load (% 1RM) [17], it is necessary to have a ds that is as low as possible and a perception of velocity as accurate as possible in order to remain within the correct training zone.

The third objective of the present study was to evaluate whether there were differences in the perception accuracy in the different intensity zones. Therefore, based on the individual load–velocity profiles obtained from the 1RM tests, for the evaluation of the PV, we chose five velocity ranges corresponding to different intensities. The delta score calculated for each load reveals that in the Pre condition, both groups were more precise at heavy loads (lower ds). The training period with the velocity feedback improved the ds for all loads, which is important in terms of training prescription. All subjects in the Pre tests overestimated the Vr in the BP. These data do not seem in line with what was reported by Bautista et al. [14] for the BP, where they reported an overestimation at low loads, an underestimation at high loads, and a higher level of accuracy at medium loads. For the SQ exercise, in the Pre condition, both groups underestimated the velocity for the heaviest loads (load 5), a finding that is in line with those found by Bautista and colleagues on the back squat [15]. However, these studies have been conducted using the Smith Machine that may reduce movement variability compared with free-weight exercises resulting in variations in barbell kinematics [18,21]. The load–velocity profile and, in particular, the minimal velocity threshold (MVT), that is, the MPV at 1RM, are exercise specific. For example, in the SQ, the MVT is about 0.30 m·s^−1^, in the Smith machine BP it is 0.15 m·s^−1^, and in the free-weight BP it is 0.18 m·s^−1^ [22,23], while in the Prone Bench Pull it is 0.50 m·s^−1^ [24,25]. For this reason, in our opinion, the use of a single PV scale for the SQ and BP could mislead subjects by increasing the error in the velocity estimation. In fact, the scale ranges from 0.1 to 1.6 m·s^−1^ and may be suitable for the BP but not for the SQ. Similarly, the “very slow” verbal anchor is located between 0.2 and 0.3 m·s^−1^, but for the SQ, it should be positioned higher, as 0.3 m·s^−1^ represents the MVT.

## 5. Conclusions

Before training, after only one familiarization session, data reveal a more precise perception of velocity in the back squat compared to the bench press. After training with the velocity feedback, the EG showed no differences in the PV accuracy between the two exercise modes. Another interesting finding was that the differences in the accuracy of the PV between the different loads leveled off after training.

This study demonstrates that training with a device that provides feedback on velocities and using the PV scale significantly increase the perception of velocity in trained subjects and thus improves the accuracy of the estimation. Therefore, athletes who regularly use this method can regulate their load based on the PV, which is specifically useful if devices are not available for athletes during all the training sessions.

## Figures and Tables

**Figure 1 jfmk-07-00034-f001:**
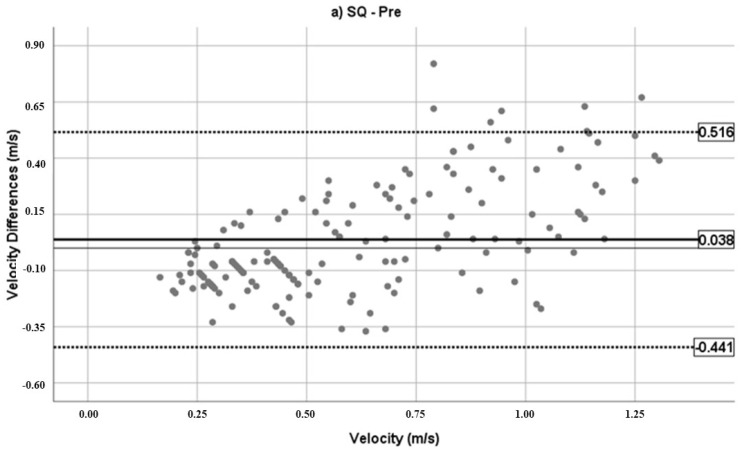
Bland–Altman plots for the agreement between real and perceived velocities. **Experimental group:** (**a**) SQ Pre; (**b**) SQ Post; (**c**) BP Pre; (**d**) BP Post. **Control group:** (**e**) SQ Pre; (**f**) SQ Post; (**g**) BP Pre; (**h**) BP Post.

**Figure 2 jfmk-07-00034-f002:**
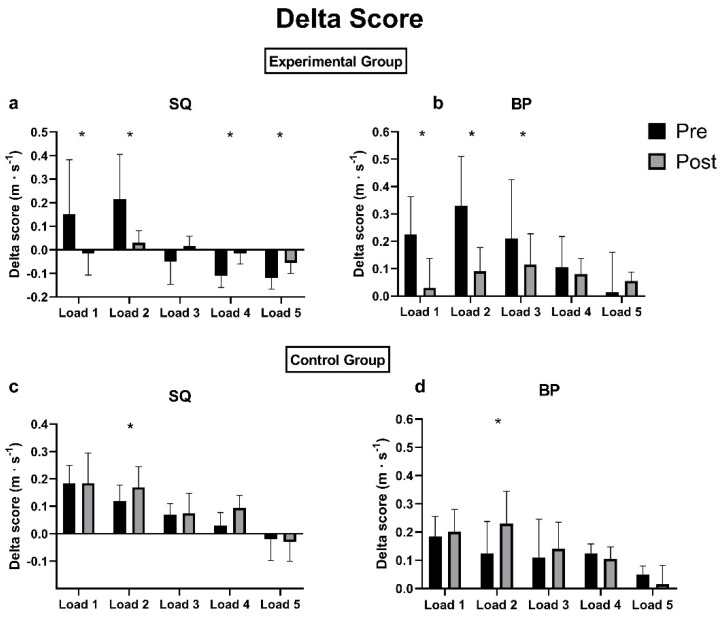
Delta score for each load from the lightest (load 1) to the heaviest (load 5). **Experimental group:** (**a**) SQ; (**b**) BP. **Control group:** (**c**) SQ; (**d**) BP. * *p* < 0.05.

**Table 1 jfmk-07-00034-t001:** Anthropometric characteristics and strength levels (mean ± SD). EG = experimental group; CG = control group.

Group		Age (Years)	Body Mass (kg)	Height (cm)	Back Squat		Bench Press	
					1RM (kg)	1RM/BW	1RM (kg)	1RM/BW
**EG**	Males	26.9 (±5)	82.2 (±7.5)	183.3 (±5.4)	118.3 (±20.5)	1.44 (±0.24)	87.8 (±13.5)	1.07 (±0.16)
	Females	26 (±3.8)	60 (±7.5)	169.8 (±3.6)	96.4 (±18.2)	1.61 (±0.28)	64.5 (±15.5)	1.07 (±0.24)
**CG**	Males	26.9 (±3.2)	79.6 (±15.5)	178.8 (±9.4)	139.6 (±23.3)	1.79 (±0.33)	93.2 (±18.6)	1.20 (±0.26)
	Females	23.8 (±2.9)	59.8 (±5.6)	164.5 (±3.7)	86.4 (±14.5)	1.45 (±0.25)	59.5 (±11.7)	1.0 (±0.15)

**Table 2 jfmk-07-00034-t002:** Maximal incremental test protocol.

1RM TEST (Back Squat—Bench Press)			
*Protocol*			
MPV (m·s^−1^)	Load	Reps	Rest (min)
	20 kg	5	3
>0.8	+20 kg	3	3
0.6–0.8	+10 kg	2	3
0.5–0.6	+5 kg	1	3
<0.5	+2.5 kg	1	3

**Table 3 jfmk-07-00034-t003:** Intensity intervals for the choice of loads in the PV assessment.

Blinded Load Test	
*Protocol*	
Exercise	MPV (m·s^−1^)
Bench Press	>1
	0.6–0.8
	0.5–0.6
	0.3–0.5
	<0.3
Back Squat	1–1.2
	0.7–0.9
	0.5–0.6
	0.4–0.5
	<0.4

**Table 4 jfmk-07-00034-t004:** Training sessions per week. ***RIR***
*= repetitions in reserve*.

*Session*	*Exercise*	*Sets*	*Reps*	*RIR*	*Rest*
* **1** *	*Back Squat*	1	10	2–3	3 min
		1	8	2–3	
		2	6	1	
		2	4	1	
		2	2	1	
	*Bench Press*	1	15	2–3	3 min
		1	12	2–3	
		2	10	1	
		2	8	1	
* **2** *	*Bench Press*	1	10	2–3	3 min
		1	8	2–3	
		2	6	1	
		2	4	1	
		2	2	1	
	*Back Squat*	1	15	2–3	3 min
		1	12	2–3	
		2	10	1	
		2	8	1	

**Table 5 jfmk-07-00034-t005:** Delta Score = Perceived velocity − Real velocity.

DELTA SCORE (m·s^−1^)	*Group*	
	EG	CG
	Median ± Interquartile Range	Median ± Interquartile Range
* **SQ Pre** *	−0.035 ± 0.35	−0.055 ± 0.18
* **SQ Post** *	−0.010 ± 0.14	−0.080 ± 0.20
* **BP Pre** *	−0.190 ± 0.29 *	−0.105 ± 0.18
* **BP Post** *	−0.070 ± 0.15 *	−0.130 ± 0.21

**EG** = experimental group; **CG** = control group. *****
*p* < 0.05.

## Data Availability

Data available on request due to restrictions (privacy).

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
