# Peer review of "Perception of Velocity during Free-Weight Exercises: Difference between Back Squat and Bench Press"

_jfmk, 2022, doi:10.3390/jfmk7020034_

Round 1

Reviewer 1 Report

This is a well written paper which investigated the possibility of improving perceived velocity during squat and bench press after a period of training. The authors have provided clear justification for the study, appropriate study protocol, clear reporting and interpretation of the results.

There are only two minor amendments that i hope the authors would consider working on.

Line 140 – 146 & Table 4: Please state the intensity of the load lifted. Was it based on % 1RM or velocity? Did participants have to lift to failure or to certain velocity threshold?

Line 161-166: Please include the standard deviation for the percentages.

Author Response

  • This is a well written paper which investigated the possibility of improving perceived velocity during squat and bench press after a period of training. The authors have provided clear justification for the study, appropriate study protocol, clear reporting and interpretation of the results.

We thank the reviewer for the appreciation.

  • There are only two minor amendments that i hope the authors would consider working on.
  • Line 140 – 146 & Table 4: Please state the intensity of the load lifted. Was it based on % 1RM or velocity? Did participants have to lift to failure or to certain velocity threshold?

Answer: We thank the reviewer for the comment. During the training program we prescribed sets and reps while the participants independently chose the load, based on repetitions-in-reserve (RIR) as already reported by the following papers. [Balsalobre-Fernández, C.; Muñoz-López, M.; Marchante, D.; García-Ramos, A. Repetitions in Reserve and Rate of Perceived Exertion Increase the Prediction Capabilities of the Load-Velocity Relationship. J. Strength Cond. Res. 2018] [Zourdos, M.C.; Klemp, A.; Dolan, C.; Quiles, J.M.; Schau, K.A.; Jo, E.; Helms, E.; Esgro, B.; Duncan, S.; Garcia Merino, S.; et al. Novel Resistance Training–Specific Rating of Perceived Exertion Scale Measuring Repetitions in Reserve. J. Strength Cond. Res. 2016, 30, 267–275]. We have included the explanation in lines 140-143 and in Table 4.

  • Line 161-166: Please include the standard deviation for the percentages.

Answer: The percentages in question represent the count of "yes" and "no" of the total group. Each subject for each of the 5 loads had to indicate the fastest repetition of the two performed. "Yes" when the subject correctly identified the fastest repetition, "no" if he was wrong. For example, in the blinded load test in the Squat exercise, in the Pre condition, the subjects of the Experimental Group were 16, so the total cases were 80 (16x5). By adding the "Yes" of each participant, we obtained 44 correct answers, out of a total of 80 (55%).

Reviewer 2 Report

The authors are demonstrating that the bar velocity improves with specific training and that differences in the accuracy in loads and exercise modes seen prior to training are leveled off after training. Although the results are interesting and it are going to help coach to set up a best training, the methods and results are not well written.

1) Materials and methods:

1.1) In table 1:  Please add more information to anthropometric characteristics such as weight, height and also strength levels of men and women separately.

1.2) It was unclear whether the same maximal incremental testing protocol was also applied to women, and if so, please describe why.

1.3) Was the blind load test also applied to the control group?

2) Results:

2.1) Figure 1: Add the letters of each graph in the upper left corner and also increase the numbers of each graph. It is impossible to read the information. Please try to add the information at the top of the graph for each group the dashboard is related to.

2.2) Delta score (ds): write in more detail the meaning of Vp and Vr and also the reason why the authors used this calculation before describing the results. This information is in the discussion section, I believe it would be more useful in the results section.

Author Response

  • The authors are demonstrating that the bar velocity improves with specific training and that differences in the accuracy in loads and exercise modes seen prior to training are leveled off after training. Although the results are interesting and it are going to help coach to set up a best training, the methods and results are not well written.
  • We thank the reviewer for the time dedicated in reviewing our manuscript
  •  
  • 1) Materials and methods:
  • 1) In table 1: Please add more information to anthropometric characteristics such as weight, height and also strength levels of men and women separately.

Answer: We thank the reviewer for the comment and we have added the required information.

  • 2) It was unclear whether the same maximal incremental testing protocol was also applied to women, and if so, please describe why.

Answer: Yes, all the participants performed the 1RM test in the same way. When the MPV was less than 0.5 m · s−1, the load was increased in smaller increments until 1RM was reached. Considering the 1RM as the heaviest load that subjects could lift correctly. Similar protocols have been used in other studies [Caven, E.J.G.; Bryan, T.J.E.; Dingley, A.F.; Drury, B.; Garcia-Ramos, A.; Perez-Castilla, A.; Arede, J.; Fernandes, J.F.T. Group versus Individualised Minimum Velocity Thresholds in the Prediction of Maximal Strength in Trained Female Athletes. Int. J. Environ. Res. Public Health 2020, 17, 7811] [Fernandez Ortega, J.A.; los Reyes, Y.G. De; Garavito Peña, F.R. Effects of strength training based on velocity versus traditional training on muscle mass, neuromuscular activation, and indicators of maximal power and strength in girls soccer players. Apunt. Sport. Med. 2020, 55, 53–61].

  • 3) Was the blind load test also applied to the control group?

Answer: Yes, the blinded load test was performed by both groups (line 128-129). This test was necessary to verify the effects of training the perceived velocity

  • 2) Results:
  • 1) Figure 1: Add the letters of each graph in the upper left corner and also increase the numbers of each graph. It is impossible to read the information. Please try to add the information at the top of the graph for each group the dashboard is related to.

Answer: We thank the reviewer for the comment, the figure has been amended accordingly

  • 2) Delta score (ds): write in more detail the meaning of Vp and Vr and also the reason why the authors used this calculation before describing the results. This information is in the discussion section, I believe it would be more useful in the results section.

Answer: We thank the reviewer for the comment, and included a sentence also in the results to help the reader get acquainted with this score. As already reported by Bautista et al. [Bautista, I. J., Chirosa, I. J., Robinson, J. E., Chirosa, L. J., & Martínez, I. (2016). Concurrent validity of a velocity perception scale to monitor back squat exercise intensity in young skiers. The Journal of Strength & Conditioning Research, 30(2), 421-429], we calculated the difference between the Perceived Velocity (Vp) and the Real Velocity (Vr). In this way, the value obtained provides quick and intuitive information on the level of accuracy of the subject.

Round 2

Reviewer 2 Report

I would like to thank the authors for accepting my suggestion and for making the changes. The latest version of the article is improved and well done. I have no more comments.

Thanks again for the opportunity to review this article.